# Pediatric Glaucoma—From Screening, Early Detection to Management

**DOI:** 10.3390/children10020181

**Published:** 2023-01-18

**Authors:** Ruyue Shen, Venice S. W. Li, Mandy O. M. Wong, Poemen P. M. Chan

**Affiliations:** 1Department of Ophthalmology and Visual Sciences, The Chinese University of Hong Kong, Hong Kong, China; 2Hong Kong Eye Hospital, Kowloon, Hong Kong, China; 3Lam Kin Chung. Jet King-Shing Ho Glaucoma Treatment and Research Centre, Department of Ophthalmology and Visual Sciences, The Chinese University of Hong Kong, Hong Kong, China; 4Department of Ophthalmology and Visual Sciences, The Prince of Wales Hospital, Hong Kong, China

**Keywords:** pediatric glaucoma, etiology, screening, diagnosis, treatment

## Abstract

Pediatric glaucoma (PG) covers a rare and heterogeneous group of diseases with variable causes and presentations. Delayed diagnosis of PG could lead to blindness, bringing emotional and psychological burdens to patients’ caregivers. Recent genetic studies identified novel causative genes, which may provide new insight into the etiology of PG. More effective screening strategies could be beneficial for timely diagnosis and treatment. New findings on clinical characteristics and the latest examination instruments have provided additional evidence for diagnosing PG. In addition to IOP-lowering therapy, managing concomitant amblyopia and other associated ocular pathologies is essential to achieve a better visual outcome. Surgical treatment is usually required although medication is often used before surgery. These include angle surgeries, filtering surgeries, minimally invasive glaucoma surgeries, cyclophotocoagulation, and deep sclerectomy. Several advanced surgical therapies have been developed to increase success rates and decrease postoperative complications. Here, we review the classification and diagnosis, etiology, screening, clinical characteristics, examinations, and management of PG.

## 1. Introduction

Pediatric glaucoma (PG), also referred to as “childhood glaucoma” or “congenital glaucoma”, covers a heterogeneous array of diseases. It is characterized by ocular structural damage and visual impairment associated with increased intraocular pressure (IOP) [1]. According to the World Glaucoma Association (WGA) consensus, PG is classified as primary or secondary, in which primary PG can be further divided into primary congenital glaucoma (PCG) and juvenile open-angle glaucoma (JOAG) [2]. Pediatric glaucoma can lead to irreversible blindness and bring emotional or psychological burdens to patients’ caregivers [3,4]. The incidence of pediatric glaucoma varies across different populations. For instance, the estimated annual incidence rate of pediatric glaucoma among patients aged <20 years on presentation is 0.92 per 100,000 populations in Hong Kong [5]. In a series in the United States (US), the reported incidence was 2.29 per 100,000, or 1 per 43,575 residents younger than 20 years [6]. A higher incidence was found in Slovakian gypsies (1/1250) [7] and in Saudi Arabia (1/2500) [8].

PG has various presentations and subtle symptoms, making the diagnosis difficult. A high index of suspicion by pediatricians or family physicians and prompt referral to an ophthalmologist for a full ophthalmic examination is essential in the timely diagnosis of pediatric glaucoma. Effective management of lowering IOP is warranted to prevent visual loss and achieve a better visual prognosis. Visual prognosis depends on the initial disease presentation, promptness of interventions, and the extent of structural damage at presentation (e.g., the extent of optic nerve damage, corneal damage, amblyopia, and progressive refractive errors). This review aims to summarize current evidence regarding classification and diagnosis, etiology, screening, clinical characteristics, examinations, and management of PG, and to discuss future challenges and directions in the PG management.

## 2. Classification and Diagnosis

A standardized, practical classification system is essential for physicians and researchers to better understand the prevalence, clinical characteristics, and diagnosis, thereby improving the management of pediatric glaucoma. Over the past decades, terms used to describe this condition include “developmental”, “infantile”, or “congenital”, which are inconsistent and ambiguous. Although several classification systems have been developed for differential diagnosis of primary and secondary childhood glaucoma, no unified criteria are used in the clinic and research registries. Recently, the Childhood Glaucoma Research Network (CGRN) proposed a systematic, reproducible classification system for childhood glaucoma, which has become the first International Consensus Classification [9,10]. The CGRN classified glaucoma children into three categories: glaucoma suspect (GS), primary glaucomas (PCG and JOAG), and secondary glaucomas (glaucoma following cataract surgery [GFCS], glaucoma associated with non-acquired systemic disease or syndrome, glaucoma associated with non-acquired ocular anomalies, glaucoma associated with acquired conditions). The CGRN further developed a flowchart to illustrate the classification system in a logical manner (Figure 1). 

To date, the CGRN classification has been successfully applied to describe the etiology, prevalence, clinical features, treatment, and outcomes of childhood glaucoma among worldwide populations. Lopes et al. used CGRN classification in the childhood glaucoma clinic of a Brazilian tertiary care center to retrospectively review the medical records and found that PCG was the most common subtype (43.95%). JOAG was the less common subtype (0.40%) [11]. Similarly, using the CGRN criteria to reclassify patients, PCG was the most common childhood glaucoma in India (32%) [10] and Malawi (93.3%) [12]. In Boston, the most common diagnosis was GFCS (36.5%), followed by PCG (29.0%) [13]. Of note, many patients were reclassified when the CGRN criteria were applied in the studies mentioned above because CGRN has stricter standards to define glaucoma suspects and glaucoma. For instance, Tam et al. recategorized several patients initially coded as glaucoma to GS because only one CGRN criterion for glaucoma was met. Many patients initially diagnosed as GS were excluded due to physiologic cupping of the optic disc or non-reproducible increased IOP [13]. Overall, a clear, unified classification system is highly suggested for future studies on childhood glaucoma or glaucoma suspect.

## 3. Etiology

The molecular etiology of PCG has yet to be fully understood. Recent genetic studies have improved our understanding of the role of genetics and the heterogenic pathophysiological pattern of PCG. PCG is usually inherited in an autosomal recessive pattern. High prevalence of PCG was found among consanguineous marriages [14]. The most reported mutated gene in PCG is *CYP1B1* (cytochrome P450 family 1 subfamily B member 1). The exact function of *CYP1B1* in the development of the eye remains unclear. However, mutations in this gene are believed to be associated with structural defects in the trabecular meshwork and the aqueous humor outflow pathways [14,15]. Up till now, several sequence variations and missense mutations in the *CYP1B1* gene have been identified in PCG probands born to parents who have consanguineous marriages in Pakistan [16,17,18]. In Spain, nearly 30% of PCG patients carry loss-of-function *CYP1B1* variants, most of which result in null genotypes [19]. The genotype–phenotype correlations varied according to different populations [20] and ethnic groups [21]. *CYP1B1* mutation was associated with a higher degree of postoperative haze [22], earlier onset disease [23,24], more severe manifestations, and more severe prognosis [23,24]. Additionally, a large Moroccan cohort reported that *CYP1B1* mutations were associated with a more severe prognosis [25], indicating its potential clinical application in predicting disease prognosis.

Another well-known causative gene is *LTBP2* (latent transforming growth factor-beta binding protein 2). LTBP2 protein is expressed in the trabecular meshwork and has an important role in aqueous humor production, IOP regulation, and ciliary zonule development [26]. Missense and frameshift have been identified in *LTBP2* in PCG families and patients, indicating that mutations in the *LTBP2* gene are possible causes of ocular anomalies that may cause IOP elevation and eventually lead to PCG [27,28,29]. 

In addition to *CYP1B1* and *LTBP2*, *MYOC* (Myocilin) [30], *TEK* (the tunica intima endothelial receptor tyrosine kinase) [31], *CPAMD8* [32], *PLOD2* (procollagen-lysine 2-oxoglutarate 5-dioxygenase 2) [33], *GPATCH* [34], and *PRSS56* [35] were also found to be associated with PCG. Apart from PCG, some secondary pediatric glaucomas were also associated with gene mutations. For example, mutations in *FOXC1* (forkhead box transcription factor C1) and *PITX2* (paired-like homeodomain transcription factor 2) were found in Axenfeld–Rieger syndrome [36,37]; genotype–phenotype mutations in *PAX6* (paired box 6) were found to be associated with aniridia [38]. However, the function of these genes has not yet been fully identified. Furthermore, currently identified genes only explain 5–50% of those affected patients [29,30,31,32,39,40]. Ongoing studies are needed to focus on novel mutations and their pathogenic function identifications. This will provide more information on genetic counseling, screening, and potential gene therapy.

## 4. Screening

### 4.1. Importance of Awareness of Caregivers and Clinicians

Pediatric glaucoma accounts for 10.8% of visual impairment in children [41] and congenital glaucoma accounts for 4.2% of childhood blindness [42]. The visual prognosis of these patients depends on the timing of presentation and treatment. Delay in presentation, diagnosis, and management can lead to devastating visual outcomes. The delayed presentation is mainly due to a lack of awareness of the disease among caregivers and clinicians. A large tertiary center study in South India showed that nearly half of PCG children had delayed presentation to the tertiary center by greater than three months from the time that caregivers recognized symptoms, although most parents or caregivers identified symptoms within the first week after birth [43]. The reasons for delayed presentations in this setting include relatively poor socioeconomic status, limited access to healthcare, and long travel time to the tertiary center. It is essential for healthcare policy, especially in developing countries or low-resource regions, to implement educational efforts to improve caregivers’ awareness regarding the need for early intervention for PCG and to ensure the early detection of the disease. 

### 4.2. Patterns of Referral

There is a need to improve the efficiency of referring childhood glaucoma patients to specialist centers. A recent survey in Brazil reported that among those patients who were referred to a pediatric glaucoma center, glaucoma was confirmed in 49% of patients. However, the diagnosis of glaucoma was ruled out in 25% of referred glaucoma patients, and 25% of glaucoma patients were diagnosed as suspected glaucoma for continuing outpatient follow-up [44]. Elevated IOP (>21 mmHg) and external abnormalities (e.g., corneal opacity, enlarged eyeballs, tearing, and photophobia) were the main referral reasons. Despite this, healthcare professionals or general ophthalmologists should remain vigilant in evaluating relevant signs, especially for children with visual anomalies or systematic diseases known to be associated with glaucoma, such as uveitis, aniridia, aphakia, pseudophakia, Sturge–Weber syndrome (SWS), Marfan syndrome, and Weill–Marchesani syndrome (WMS) [45]. 

Appropriate efforts should also be made towards an effective referral pattern for underserved regions or less resourceful areas because glaucoma children who live in these regions suffer the additional burdens of limited access to specialist care. Recently, a geospatial service coverage analysis was applied to identify the number of infants at risk of delayed PCG evaluation due to long travel distances or time to specialists [46]. This cross-sectional analysis was performed by geocoding all American Glaucoma Society (AGS) and American Association for Pediatric Ophthalmology and Strabismus (AAPOS) provider locations to generate 1-h drive time areas to providers and overlaying these regions with demographic data. The service coverage analysis estimated that approximately 14 to 94 new PCG cases per year are at risk of delayed diagnosis due to their remote locations with shallow service coverage areas [46]. These data may help improve the efficiency of PCG screening strategies.

## 5. Clinical Characteristics 

### 5.1. Anterior Segment Abnormalities 

Increased corneal diameter and buphthalmos are significant abnormality to be recognized in pediatric glaucoma and usually occurs in young children before the age of three [2] (Figure 2). The Haab striae (i.e., breaks in the Descemet membrane [DM]) were noted in 44.8% of eyes and were most frequently present between 3 and 5 mm from the optical axis [47]. Corneal clouding is a risk factor for blindness among pediatric glaucoma patients [48]. Recently, abnormal corneal irregularity and corneal high-order aberrations (HOAs) were reported among over 60% of PCG eyes [49]. More importantly, the abnormalities of corneal configurations and increased HOAs are related to degraded visual outcomes. Adequate evaluation of the cornea, appropriate aberration correction, and amblyopia treatment are essential to improve visual outcomes. 

Other corneal findings on PCG include a significant decrease in corneal hysteresis (CH), corneal resistance factors (CRF) and central corneal thickness (CCT) [50,51], lower cell density of endothelial cells [52], and thickening of DM and pre-Descemet layer (PDL) [53]. Future studies are needed to understand these corneal findings’ significance and possible long-term consequences.

While isolated trabeculodysgenesis is known to contribute to PCG pathogenesis [54], the iris may appear normal or associated with stromal hypoplasia, loss of iris crypts, peripheral scalloping of posterior pigment iris layer and prominent iris vessels [2]. PCG eyes were also reported to have a greater distance between the anterior edge of the ciliary body (CBD) and the edge of the corneoscleral limbus. The knowledge regarding the greater CBD and its variability in PCG eyes could enable better planning of surgical treatment for those patients [55]. Attention should also be drawn to features of concomitant ocular disorders (see the section below). 

A careful anterior segment examination may also influence management plans. For example, concomitant cataract may prompt the surgeon to incline toward glaucoma drainage device implantation for better control after lensectomy [2]. It was also reported that congenital nasolacrimal duct obstruction (CNLDO) occurred in 2.5% of congenital glaucoma patients among patients 1-year-old or less. Prompt treatment of CNLDO was suggested to prevent sight-threatening ocular infection with CNLDO and to minimize delay to glaucoma surgery [56].

### 5.2. Posterior Segment Changes

Assessment of the optic nerve head (ONH) is crucial for the diagnosis and monitoring of pediatric glaucoma. Abnormalities of the optic disc, including increasing or increased cup–disc ratio (CDR), CDR asymmetry, and focal thinning, form one of the criteria for defining glaucoma or glaucoma suspect [2]. Of note, CDR parameters (i.e., CDR enlargement or asymmetry) had a high false-positive rate as a referral sign; only 8.5% of children referred with enlarged CDR or CDR asymmetry had confirmed glaucoma [44]. The appearance of optic disc (i.e., ONH cupping reversal) can be improved after considerable IOP reduction in pediatric glaucoma [57,58]. A higher prevalence of ONH cupping reversal was found in younger eyes [59]. However, some eyes with ONH cupping reversal still experienced continual disease progression after IOP-lowering surgery. This reflects that cupping reversal in pediatric glaucoma may not predict the improvement of the ONH status [58].

### 5.3. Change in Axial Length and Refraction

Axial elongation has been found in pediatric glaucoma. Increase of the axial length leads to secondary high axial myopia, with thinning of the sclera anterior and posterior to the equator. The scleral and choroidal thinning in myopia may be due to a rearrangement of tissue and not due to the new formation of tissue [60]. Given the potential visual impairment caused by axial elongation, management of axial elongation and high myopia should also be considered as one of the important sessions for PG management.

### 5.4. Characteristics of Secondary Glaucomas

The clinical characteristics of secondary glaucomas vary due to different initial causes. For example, ectopia lentis can be present in glaucoma patients secondary to WMS [45] or Marfan syndrome [61]. Port-wine stains (PWS) involving the eyelids are more likely to be associated with ipsilateral glaucoma. Two series conducted in South Korea and the United Kingdom showed significant association with glaucoma with lower eyelids and combined upper and lower eyelid PWS, respectively [62,63]. Congenital infection should also be considered in neonates with glaucoma, as neonatal-onset glaucoma is an essential component of congenital rubella syndrome (CRS), which may present without buphthalmos and persistent corneal clouding despite reasonable IOP control. In an Indian series of 27 infants with neonatal-onset glaucoma, 25.9% of those with newborn glaucoma had underlying intrauterine rubella infection [64]. 

## 6. Examinations

Examination under anesthesia (EUA) is usually required for a definite evaluation of infants and young children with suspected glaucoma. Gonioscopy is essential for angle assessment, angle status observation, glaucoma-type classification, and management. Measurement of pre-anesthesia IOP, refraction, pachymetry, axial length, and corneal diameter should be performed. If pre-anesthetic IOP measurement is not possible, IOP should be assessed first to limit the confounding effect of general anesthesia [65,66,67,68]. With the development of ocular imaging technology, several novel instruments have emerged to be used in clinical evaluation for PG. 

### 6.1. IOP Measurement

Measurement of IOP in children is challenging because younger children are usually less cooperative with eye examinations. Goldmann applanation tonometry (GAT) is the gold standard of IOP measurement, but it is usually difficult for pediatric patients. Other handheld applanation tonometry, such as Perkins tonometer, Tono-Pen, and rebound tonometer, could be helpful. A retrospective analysis of IOP measured by noncontact tonometry, rebound tonometry, and GAT in 419 children aged 3 to 15 years old found that for children less than 10 years old, it was most likely to measure the IOP successfully using noncontact tonometry, followed by rebound tonometry and GAT [69]. There was a higher success rate in children older than 10 years old in all three types of tonometry compared to children younger than 10 [69]. However, both GAT and handheld applanation tonometry require anesthetic eye drops and sustained cornea contact, which may not be tolerated by younger children, rendering accurate IOP measurement difficult. Rebound tonometry (iCare, iCarePro [ICP]) does not require anesthetic eye drops and allows assessment of young infants or patients who cannot sit upright [70]. ICP can estimate IOP reasonably in selected children whose IOP cannot be obtained by other means of measurement. Although ICP had a higher success rate than GAT in IOP measurement [71], IOP measured by iCare tends to be higher than that measured by GAT in the same setting [72,73,74,75,76,77]. For the population of supine sedated children with glaucoma, IOP measurements with ICP tend to be lower than readings from the pneumotonometer and Tono-Pen [78,79].

### 6.2. Coreanl Diameter Measurement

The routinely used device to measure corneal diameter in PCG is the Castroviejo caliper [80,81]. Since contact is required with the caliper’s pointed arms, the examination is usually performed under anesthesia. Recently, Bafna et al. devised a simple U-shaped tool to make it more feasible to screen congenital glaucoma by first-contact physicians or optometrists [82]. The U-tool was constructed by three Schirmer strips glued to each other with a “U” shape. It was placed at the level of the orbital rims or just above it. The results measured by U-tool showed a good correlation with the value measured by a Castroviejo caliper [82]. Since it is a non-contact tool, it can also be used by ophthalmologists when EUA is delayed. It is also a potentially feasible and low-price tool that could be useful in screening programs. 

### 6.3. Anterior Segment Imaging

In addition to the essential gonioscopy examination, anterior segment imaging, such as ultrasound biomicroscopy (UBM) and anterior segment optical coherence tomography (ASOCT), can provide a detailed assessment of the anterior segment structures and help to guide appropriate management. UBM allows clear visualization of the anterior chamber, provides structural and functional information, allows measurement of Schlemm’s canal, and helps to identify specific secondary causes of pediatric glaucoma. For instance, Tandon et al. used UBM to measure the diameter of Schlemm’s canal and found that the mean canal diameter was smaller in glaucoma patients than in non-glaucoma subjects (64.9 µm vs. 142 µm) [83]. UBM assessment in PCG showed abnormal insertion of the iris and ciliary body, thinner iris, wider trabecular-iris angle (TIA), thinner ciliary body and lens thickness, or absence of the Schlemm’s canal, and presence of abnormal tissue membrane covering the trabecular meshwork. UBM imaging provides additional information on anterior chamber angle dysgenesis and might help in more precise phenotyping and in choosing the best surgical option for a patient [84]. 

UBM is a contact investigation for examining the angle configuration and ciliary body’s status. It is operator-dependent, time-consuming, and lacks standardized normative values, while interpreting the results can be challenging. In order to make UBM image analysis more feasible, Alexander et al. developed a novel, open-access image analysis tool called EyeMark, for semi-automated analysis of UBM images [85]. Semi-automation may hopefully expand the use of Quantitative-UBM (Q-UBM) imaging to predominantly qualitative purposes. 

Anterior segment optical coherence tomography (ASOCT) is another imaging technology for a comprehensive view of the anterior chamber [86]. With advances in OCT technology, in-depth morphologic analysis of angle from ASOCT imaging with higher resolution may be possible and be applied to guide choices of surgical modalities. However, tabletop ASOCT imaging can be challenging for uncooperative pediatric patients and cannot image the ciliary body. Further developments in technology, such as intraoperative microscope-integrated OCT, are expected to make pediatric examination OCT imaging more feasible [87].

### 6.4. Optical Coherence Tomography (OCT) Assessment

OCT has become an essential tool for assessing retinal nerve fiber layer thickness (RNFL), with a satisfactory short-term reproducibility [88] and longitudinal reproducibility in pediatric glaucoma [89,90]. However, these studies have small sample sizes and limited longitudinal follow-up time. Further investigation involving a population of a larger number of children and possibly a subgroup who show true clinical progress would allow for better assessment of the reproducibility and analysis of sensitivity and specificity of a change if detected.

Apart from optic disc OCT assessment, macular OCT has additional value in evaluating retinal layer thickness and visualizing macular morphology [91]. For example, macular OCT can be helpful in measuring retinal thickness if the morphology of the optic nerve head is altered by high myopia and axial elongation [91,92]. Additionally, macular OCT measurement on ganglion cell layer (GCL), inner plexiform layer (IPL), and RNFL reported a high diagnostic accuracy and sensitivity, reflecting its ability to identify glaucomatous eyes and may play a role in glaucoma screening [91]. Therefore, a combination of both optic nerve head and macular segment assessment is recommended in screening and diagnosing PG.

To facilitate OCT assessment in pediatric patients, attempts have been made to establish a normative database of RNFL and ganglion cell-inner plexiform layer (GCIPL) thickness in children. A well-defined normal distribution of RNFL thickness in children could facilitate the investigation and management of pediatric glaucoma. Rao et al. analyzed 148 eyes of 74 children < 18 years, the mean RNFL thickness was 94 ± 10.9 µm and 93 ± 10.6 µm in the right and left eyes, respectively, with maximum thickness found in the superior quadrant [93]. The RNFL thickness decreases in children as myopic shift or axial length elongation. Accordingly, Goh et al. established a normative macular GCIPL and RNFL thickness in children with refractive errors [94]. In 243 eyes of 139 children, the mean spherical equivalent refraction was −3.20 ± 3.51 D, and the mean AL was 24.39 ± 1.72 mm; the mean average RNFL thickness was 99.00 ± 11.45 µm, the average GCIPL thickness was 82.59 ± 6.29 µm [94]. The normative database of RNFL and GCIPL thickness in emmetropic or myopic children may assist in evaluating disease progression and treatment efficiency for PG children.

In view of the poor cooperation of children during tabletop OCT imaging, more feasible OCT modalities have emerged, including handheld OCT [95,96] and overhead-mounted OCT [97]. Handheld OCT is feasible in PCG patients without anesthesia or sedation, particularly for children younger than 4–5 years [98]. Overhead-mounted OCT has been successfully applied to measure the optic nerve and macular thickness in glaucoma children unable to cooperate with tabletop OCT [97]. However, the image quality and scanning speed still need to be improved. Technology enhancement may help to overcome the challenge.

### 6.5. Retinal Imaging

The RetCam fundus image is designed to obtain wide-field photographs of the fundus and has been used in glaucoma management to image the optic disc and the anterior chamber angle [99]. RetCam provides a novel method to investigate optic disc morphology in infants and uncooperative children although the instrument can be costly in developing countries. MII RetCam assisted smartphone-based fundus imaging (MSFI) has been successfully utilized in fundus imaging in the pediatric age group [100]. The images were captured by a smartphone camera and then transferred for storage and management by the in-built app. MSFI has become a potential tool in low-resource areas for monitoring retinopathy of prematurity [101,102]; it is worthwhile to explore the potential role of MSFI in monitoring and detecting glaucoma in children. 

## 7. General Principles of Management

Apart from controlling the IOP, the management of concomitant amblyopia and other associated ocular pathologies are equally crucial in enhancing the visual outcome of pediatric glaucoma patients. Lifelong follow-up and proper management for postoperative complications are often needed throughout the life course of patients.

IOP-lowering treatment in pediatric glaucoma ranges from surgery to medication and laser, particularly cyclophotocoagulation. Much has evolved over the last two decades, especially in surgical techniques and devices available. The treatment choice depends on glaucoma subtypes and the status of the anterior chamber angle. For example, angle surgery is usually the first-line treatment to be considered in PCG. At the same time, medication is often used before surgical treatment in JOAG and secondary glaucoma with open angles. Glaucoma surgery is generally regarded as more challenging than adult cases due to the distorted anatomy in buphthalmic eyes, more aggressive inflammatory and healing response, and lack of postoperative cooperation for monitoring in children. Collaboration between glaucoma specialist, pediatric ophthalmologists, and caretakers are of utmost importance [2].

## 8. Surgical Treatments

Surgical treatments can be classified into procedures that improve the physiological aqueous outflow drainage (i.e., angle surgery), that create an alternative aqueous drainage pathway (i.e., trabeculectomy and glaucoma drainage device [GDD]), and that reduce aqueous production by ciliary body destruction (i.e., cyclophotocoagulation). The selection of surgical treatment depends on the types of glaucoma, the anatomy (e.g., corneal clarity), the site of conjunctival scarring, the target IOP, and the surgeon’s training and experience. Several clinical studies have been conducted to compare the safety and efficacy of different surgical treatments, including prospective randomized clinical trials. We summarized these comparative studies in Table 1. 

### 8.1. Angle Surgeries 

#### 8.1.1. Goniotomy

In 1942, Barkan first reported an operation in congenital glaucoma by adding a lens to visualize the chamber angle during surgery when performing the incision of the trabecular meshwork. This surgical procedure was named “goniotomy” [115]. In 1949, Scheie further demonstrated that goniotomy could be successfully used for treating congenital glaucoma, and IOP was not elevated even after 2 years of operation [116]. More recent studies reported that goniotomy is effective in lowering IOP, with a success rate of 30–93.5% [117,118,119]. With a clear view, goniotomy is usually safe, effective, and easy to perform. It also has the advantage of avoiding bleb-related complications and leaves no foreign body inside the eye. An endoscopic goniotomy can be performed if the patient’s cornea is not clear enough for a safe goniotomy. Goniotomy has been utilized as first-line surgery for PCG, especially for mild PCG, with a success rate of 81–100% for mild PCG [120]. Goniotomy also showed an overall success rate of 75% for young patients with refractory glaucoma secondary to chronic childhood-onset uveitis [121]. However, factors associated with the failure of goniotomy should be noted. For PCG who performed goniotomy, the failure of goniotomy was associated with positive consanguinity of the parents and surgery before the end of the first month [122]. 

#### 8.1.2. Trabeculotomy

Trabeculotomy is another alternative angle surgery as a first-line treatment for PCG, particularly in older patients [123,124]. Trabeculotomy was first introduced in 1970 as an ab externo procedure [125]. Differ from goniotomy, trabeculotomy is performed by cannulating the Schlemm’s canal from an external method with subsequent centripetal rupture through the trabecular meshwork into the anterior chamber. Trabeculotomy seems to be superior to goniotomy for PCG patients, as reported by a 23-year period of follow-up study that the success rate was higher for primary trabeculotomy (78.9%) compared to primary goniotomy (20.6%) [126]. Positive consanguinity, younger age, higher preoperative IOP, and female gender are risk factors for failure of trabeculotomy [127]. 

#### 8.1.3. Improvement of Conventional Angle Surgeries

Some novel procedures and instruments have been developed to improve surgical outcomes of conventional angle surgeries. Visco-trabeculotomy (VT) is a modified probe trabeculotomy procedure in which a viscoelastic material is applied to separate tissues to prevent bleeding and fibroblastic proliferation at the stage of trabeculotomy opening. VT showed higher success rates, lower complications, and more stability than conventional probe trabeculotomy in refractory PG [110]. Newer modifications allow the angle to open 360 degrees during trabeculotomy (i.e., circumferential trabeculotomy). To combine the advantages of viscosurgical devices and circumferential trabeculotomy, visco-circumferential-suture-trabeculotomy (VCST) was introduced by Elwehidy et al. [114]. A randomized controlled study showed that VCST had a marginal advantage over VT for long-term IOP reduction at a 3-year follow-up [114]. 

Novel instruments have been developed to assist the performance of 360-degree trabeculotomy. The Trab360 device (Trab360; Sight Sciences, Menlo Park, CA, USA) is an ab interno trabeculotome instrument that allows cutting up to 360 degrees of trabecular meshwork by cannulating the SC. Trabeculotomy with Trab360 achieved a success rate of 67.4%; it was effective and safe for PG patients [128]. Microcatheter-assisted trabeculotomy (MCT) was introduced to visualize the tip of the suture. Studies that utilized MCT showed better results and significantly lower reoperation rates than conventional angle surgery [129,130].

Grover and colleagues developed gonioscopic-assisted transluminal trabeculotomy (GATT), which is an ab interno circumferential trabeculotomy approach [131]. It spares conjunctiva for future filtering surgeries, with a more significant reduction in aqueous humor outflow resistance than goniotomy alone, resulting in better IOP control [113]. GATT has been reported to be effective in JOAG and showed promising results for patients with prior filtering surgeries [132]. 

One of the complications during circumferential trabeculotomy is suture misdirection [133,134,135]. To avoid catheter misdirection, Arnav et al. [136] used indocyanine green to identify SC successfully during GATT, particularly for those eyes with poor structure differentiation. Gupta et al. [137] applied the external jugular vein (EJV) compression technique to help accurately identify SC, thus increasing the success likelihood for GATT. 

### 8.2. Filtering Surgeries

#### 8.2.1. Trabeculectomy

Trabeculectomy outcome is more favorable in patients of older age and without anterior segment anomalies [124]. Trabeculectomy success is lower than that reported in adults and varies from 35 to 50% [138]. Trabeculectomy with adjunctive MMC improved the success rate to 60 to 65% at 2 years of follow-up, although the use of antimetabolites increases the rate of complications, such as thin avascular bleb, hypotonia, and endophthalmitis [139,140]. Bleb needling with 5-fluorouracil is an efficient method for lowering IOP after a failed trabeculectomy or combined trabeculectomy and trabeculotomy in the pediatric population [141,142]. For trabeculectomy, bleb-related infection and endophthalmitis are potentially vision-threatening complications that should be promptly treated. For traumatic glaucoma, trabeculectomy with MMC effectively lowers IOP, with a success rate of 71.8% [143]. Among those eyes with failure (28.2%), the causes of surgical failure include young age and inability to control IOP immediately after surgery.

#### 8.2.2. Glaucoma Drainage Device (GDD) Surgery

The Ahmed glaucoma valve (AGV) and the Baerveldt glaucoma implant (BGI) are the two most commonly used GDDs. Other GDDs have also been reported recently [144]. GDD surgery is commonly performed particularly in secondary glaucoma, such as SWS-associated glaucoma [145], or refractory cases with failure outcome of filtering surgery [146].

AGV has satisfactory survival outcomes in eyes with pediatric keratoplasty and glaucoma [147], aphakic glaucoma [148], as well as in pediatric eyes with GFCS [149]. The use of MMC can lengthen the drop-free duration, as well as the long-term IOP control with topical medications [150]. However, AGV may later fail due to ocular growth leading to tube retraction into the corneal stroma or even completely outside the anterior chamber [151,152]. If the issue occurs, one may replace the GDD either in the same or a different location, advance the current device to a more proximal location, or extend the protruding tube with a silastic sleeve [153,154]. The use of tube extension is widely considered because it is a relatively simple procedure [153,155]. This procedure has few complications, and the efficacy of long-term outcomes (mean follow-up of 6 years) is satisfactory [156].

Although GDD is relatively safe and effective for pediatric glaucoma, the overall success rates decrease with an extension of follow-up time after surgery, ranging from 44% to 95% for a postoperative follow-up time of 1 to 6 years [157,158,159,160]. Early complications and late complications were reported. The common early complications after GDD surgery include choroidal detachment, flat anterior chamber, hyphema, and hypotony. Late complications include corneal erosion or decompensation, cataract, and phthisis bulbi [146]. One of the severe postoperative complications is the exposure of the tube or plate, as it is considered a risk factor for later onset endophthalmitis [161]. Implant exposure was associated with younger age, combined procedure at the time of primary GDD implantation, and multiple previous ocular surgeries [162]. There are some solutions for the failure cases. In refractory cases, cyclodestructive procedures such as transscleral cyclophotocoagulation and endoscopic cyclophotocoagulation could be considered. The postoperative use of eye lubricants was shown to be protective against implant exposure [162].

Several strategies have been implemented to decrease postoperative complications. Ologen (Aeon Astron Europe BV, Leiden, The Netherlands) is a biodegradable Type-I collagen matrix. It was used to reduce early postoperative scarring and to prevent the collapse of the subconjunctival space. It has been used safely in filtering surgery and increased the success and survival rates of GDD surgery [163,164,165]. As the commonly used GDDs are costly, manufacturers have tried to develop a novel, cost-effective non-valved alternative GDD [166]. The Aurolab aqueous drainage implant (AADI) is based on the design of the 350 mm^2^ BGI. A prospective randomized controlled trial showed that AADI has comparable efficacy and safety to Ahmed glaucoma valve (AGV) implant in a prospective randomized controlled trial [167]. However, the incidence of postoperative suprachoroidal hemorrhage (PSCH) among pediatric patients undergoing AADI (1.4%) was higher than adults (0.4%) [168]. Therefore, surgeons should be vigilant about the possibility of PSCH development when performing AADI in PG patients.

Management of glaucoma in pediatric eyes with corneal opacification is challenging and often requires multiple surgeries. A combined endoscopic vitrectomy and posteriorly placed GDD is a viable technique to establish aqueous humor outflow. Although the success rate is low, this surgical approach may be helpful to ultimately obtain IOP control and preserve vision in these complex eyes [169].

### 8.3. Minimally Invasive Glaucoma Surgery (MIGS)

Bleb-forming minimally invasive glaucoma surgery (MIGS) devices might be an attractive interim step for refractory childhood glaucoma before moving on to the more extensive surgical dissection such as trabeculectomy with MMC or plate-based GDDs [144]. 

Currently, a novel ab externo microshunt with MMC demonstrated promising success rates, decreased drop use, and few complications [170]. The Preserflo Microshunt (Santen, Miami, FL, USA) consists of an inert, biocompatible biomaterial called poly (styrene-block-isobutylene-block-styrene), or “SIBS,” which was originally designed to coat cardiac stents [171]. This material has been reported to reduce chronic inflammation and elicit minimal scarring [172,173].

Multiple XEN gel stents for refractory PG have been reported to give a favorable resultant IOP [174]. Larger studies with longer follow-up periods are required to determine the optimal use of XEN gel stent implantation in the pediatric population.

Since the potential complications of BGI include early hypotony and late corneal decompensation, the combination of the XEN gel stent inserted ab externo in the anterior chamber connected to the Baerveldt tube in the subconjunctival space is expected to overcome these potential complications. Therefore, a new XEN-augmented Baerveldt technique was designed in 2017 [175]. The XEN acts as a flow restrictor because of its thinner internal lumen diameter (45 μm) compared with the BGI (300 μm) [175]. This technique demonstrated a promising short-term IOP control [176,177]. The longer-term efficacy and safety require further exploration. 

### 8.4. Cyclophotocoagulation

Cyclophotocoagulation (CPC) can also be used in cases that are refractory to all medical and surgical treatments and eyes with limited visual potential. However, evidence is inconclusive of whether CPC for refractory glaucoma resulted in better outcomes and fewer complications than other glaucoma treatments [178]. In children, transscleral cyclophotocoagulation (TSCPC) reduces IOP in pediatric glaucoma secondary to SWS over a follow-up period of 3 years [179]. TSCPC was associated with a lower success rate, yet a lower complication rate as an initial intervention for secondary glaucoma compared with trabeculectomy [180]. A more novel type of TSCPC, micropulse diode laser, was proposed as a safer approach with similar effectiveness in controlling IOP in children with recurrent glaucoma [181]. Micropulse is probably a safer option for retreatment because of its lower rate of complication and results in less postoperative inflammation and pain [108]. Endoscopic cyclophotocoagulation (ECP) has also been described to treat secondary glaucoma. ECP has been shown to be an effective primary intervention for GFCS in young children in long follow-up studies [182]. Future randomized controlled trials are warranted to evaluate the safety and efficacy of cyclophotocoagulation as an initial treatment for pediatric glaucoma.

### 8.5. Deep Sclerectomy

Recently, other publications also showed that deep sclerectomy (DS) provides effective IOP reduction with less complication and shorter surgery duration than filtering surgeries in the pediatric age group [183,184,185]. A cohort of children undergoing non-penetrating deep sclerectomy showed an effective reduction in IOP and no occurrence of serious complications that are commonly seen after trabeculotomy or combined trabeculotomy–trabeculectomy [185]. There is a need for long-term follow-up data to provide more solid evidence on the safety and efficacy of DS in the management of pediatric glaucoma.

## 9. Medications in Pediatric Glaucoma

Topical IOP-lowering medications are used during the preoperative period or when sufficient IOP reduction cannot be achieved following surgery. Timolol, a topical beta-blocker, is usually the first-line agent and is generally well-tolerated in children. Timolol has a low risk of bradycardia and bronchospasm, and its risk can be lowered by performing punctual occlusion after drop application, or switching to betaxolol, a beta-1 selective antagonist, which has less effect on airways than non-selective beta-blockers. Topical carbonic anhydrase inhibitors such as brinzolamide and dorzolamide are effective alternatives with minimal side effects. Oral acetazolamide could be added for greater IOP reduction but has more systemic adverse effects such as deranged renal function, metabolic acidosis, renal stone formation, paresthesia, and confusion. Topical alpha-adrenergic agonists are contraindicated in children as they cross the blood–brain barrier and can lead to central nervous system toxicity. They are also known to cause severe side effects such as bradycardia, hypotension, hypothermia, hypotonia, apnea, and unresponsiveness in the pediatric population. Notably, alpha agonists are contraindicated in children less than 6 years old, weight < 20 kg or those with cognitive impairment in whom central nervous system suppression may go unrecognized [186]. PGAs have excellent safety profiles in children. Latanoprost is a commonly used PGA that can significantly reduce IOP in pediatric patients [187,188,189]. However, its non-response rate in children is higher than that in adults, potentially due to abnormal uveoscleral pathway in types of glaucoma that predominate in children [188]. Travoprost was another type of PGA that was found to be non-inferior to timolol in terms of lowering IOP in pediatric glaucoma patients without treatment-related systemic adverse event during a study period of three months [190]. Longer follow-up investigation may help to identify the long-term safety and efficacy of travoprost in PG patients.

### Developments and Challenges in Medical Treatments

Recently, a potent Rho kinase inhibitor, netarsudil, was introduced as IOP-lowering medication for pediatric glaucoma. Netarsudil lowers IOP by increasing uveoscleral outflow [191]. As a once-daily and first-in-class IOP-lowering medication, it is safe and effective in adults [192] and was non-inferior to timolol twice daily with tolerable ocular adverse events [193]. A retrospective case study reported that early experience with netarsudil was effective for lowering IOP in pediatric patients [194]. 

Although the IOP-lowering effect of glaucoma medications is comparable to that in the adult population, the proportion of responders seems to be significantly lower in children. Given the potentially serious systemic adverse event, measures to minimize drug absorption (e.g., using the lowest dose and the gel formulation of beta-blockers or considering the lacrimal punctum occlusion) and a more frequent follow-up schedule to ensure treatment safety should be considered in pediatric patients who are on topical glaucoma medications [195].

Low medication adherence is a crucial challenge in the long-term management of pediatric glaucoma. Barriers to medication adherence include forgetfulness, complex dosing regimen, and being too busy with other activities [196,197]. Frank discussions about the importance of medication adherence and how to prevent lapses in adherence may foster better communication between the caregivers and the treatment providers [196,197,198,199,200].

## 10. Conclusions

Delayed detection is the primary concern in the management of pediatric glaucoma. Several strategies can assist with the timely diagnosis of pediatric glaucoma. Public education for caregivers and health professionals should be strengthened. More effective screening strategies are warranted for early detection of pediatric glaucoma. Novel ocular imaging technologies can be helpful for the investigation of clinical characteristics of pediatric glaucoma eyes. Given the current challenges of surgery treatments, future studies and technologies focusing on improving surgical success rates, minimizing postoperative complications, and improving follow-up adherence are needed.

In summary, pediatric glaucoma covers a complex group of diseases and an important cause of irreversible blindness in children. The past two decades have witnessed a promising advance in improving the clinical management of pediatric glaucoma. A multidisciplinary approach with cross-specialty collaborations among ophthalmologists, pediatricians, anesthesiologists, and geneticists is necessary to tailor patient treatment. The future directions are expected to focus on reducing side effects of management, achieving a better visual prognosis, and improving the quality of life of pediatric glaucoma.

## Figures and Tables

**Figure 1 children-10-00181-f001:**
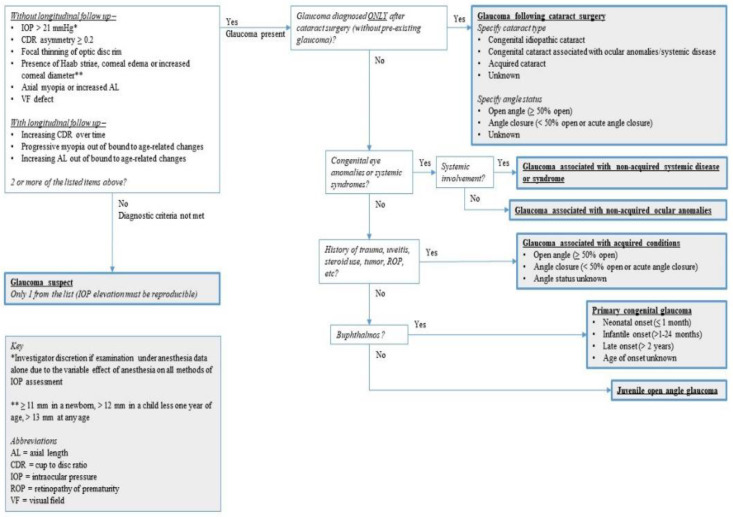
CGRN classification algorithm for pediatric glaucoma. (Figure courtesy of the Grajewski Lyra (GL) Foundation for Children with Glaucoma. Used with permission.). * Investigator discretion if examination under anesthesia data alone due to the variable effect of anesthesia on all methods of IOP assessment. ** ≥11 mm in a newborn, >12 mm in a child less one year of age, >13 mm at any age.

**Figure 2 children-10-00181-f002:**
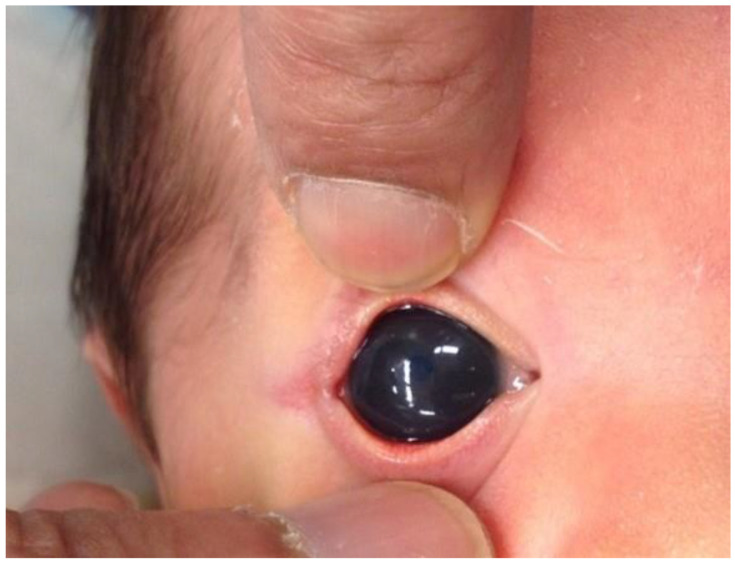
Clinical photograph of buphthalmos (right eye).

**Table 1 children-10-00181-t001:** Summary of current clinical studies on surgical treatment comparison in pediatric glaucoma.

First Author (Year)	Glaucoma Type	Study Design	Surgical Treatments	Sample Size	Main Results
Lawrence et al. (2012) [103]	PG	Retrospective, comparative	Group 1: Trabeculectomy Group 2: Combined trabeculotomy-trabeculectomy	40 eyes in 33 patients. Group 1: 17 eyes Group 2: 23 eyes	Group 2 had greater long-term success.
Eldaly et al. (2014) [104]	PCG	Prospective, comparative	Group 1: Pneumatic trabecular bypass (PTB) Group 2: Conventional trabeculotomy	42 eyes of 42 patients. Group 1: 17 eyes Group 2: 25 eyes	PTB had a greater total cumulative chance for success than group 2 (88.2% vs. 56% respectively).
Temkar et al. (2015) [105]	PCG	Prospective, randomized	Group 1: Illuminated microcatheter-assisted circumferential trabeculotomy Group 2: Combined mitomycin C-augmented trabeculotomy-trabeculectomy	60 eyes of 30 patients with bilateral PCG aged ≤ 2 years. Group 1: 30 eyes Group 2: 30 eyes	The two groups achieved comparable surgical outcomes.
Lim et al. (2015) [106]	PG	Retrospective, comparative	Group 1: 360-degree circumferential trabeculotomy Group 2: Traditional trabeculotomy (<360 degrees or partial)	91 eyes of 66 patients. Group 1: 14 eyes Group 2: 77 eyes	Group 1 had a higher surgical success rate than group 2 at 1-year (85.71% vs. 58.44%, respectively).
Shakrawal et al. (2017) [107]	PCG	Prospective, randomized	Group 1: Illuminated-Microcatheter Circumferential Trabeculotomy Group 2: Conventional partial trabeculotomy	40 eyes of 31 patients aged ≤ 2 years. Group 1: 20 eyes Group 2: 20 eyes	Group 1 performed better than group 2 at 1 year follow-up.
Abdelrahman et al. (2018) [108]	Refractory glaucoma	Prospective, comparative	Group 1: Micropulse cyclophotocoagulation Group 2: Transscleral continuous wave cyclophotocoagulation	45 eyes of 36 patients. Group 1: 17 eyes Group 2: 28 eyes	Group 1 had a higher success rate was higher (71% vs. 46% in group 2) although the difference was not significant (*p* = 0.1). Group 1 had lower rate of complications, pain, and inflammation.
El Sayed et al. (2018) [109]	PCG	Retrospective, comparative	Group 1: Microcatheter-assisted trabeculotomy Group 2: 2-site circumferential trabeculotomy using the rigid probe trabeculotome	92 eyes of 92 patients. Group 1: 33 eyes Group 2: 59 eyes	The two groups had comparable results. However, the added cost of the microcatheter in group 1 should be considered.
Elwehidy (2019) [110]	Refractory Glaucoma with failed AGV	Prospective, randomized	Group 1: Ahmed glaucoma valve revision Group 2: Visco-trabeculotomy (VT)	41 eyes of 41 patients. Group 1: 19 eyes Group 2: 22 eyes	VT had a higher success rate and a decrease in IOP-lowering medication use.
Elhofi (2020) [111]	PCG	Retrospective, comparative	Group 1: Non-penetrating deep sclerectomy Group 2: Trabeculectomy	80 eyes of 80 patients aged < 3 years. Group 1: 40 eyes Group 2: 40 eyes	Group 1 had fewer postoperative complications with a comparative postoperative IOP reduction and overall success rates.
Puthuran (2021) [112]	Refractory glaucoma	Retrospective, comparative	Group 1: Aurolab aqueous drainage implant (AADI) placed in the superotemporal quadrant Group 2: AADI placed in the inferonasal quadrant	144 eyes of 144 patients. Group 1: 96 eyes Group 2: 48 eyes	Group 1 had better IOP-related outcomes and is a safer surgical option in pediatric eyes.
Qiao (2021) [113]	Uncontrolled JOAG	Retrospective, comparative	Group 1: Gonioscopy-assisted transluminal trabeculotomy (GATT) Group 2: Kahook dual blade excisional goniotomy	46 eyes of 43 patients. Group 1: 36 eyes Group 2: 10 eyes	GATT was preferred in medical uncontrolled surgery-naïve JOAG eyes.
Elwehidy (2022) [114]	PCG	Prospective, randomized	Group 1: Visco-circumferential-suture-trabeculotomy (VCST) Group 2: Rigid probe visco-trabeculotomy	84 eyes of 49 patients Group 1: 40 eyes Group 2: 44 eyes	Group 1 provided a marginal advantage over group 2.

**Abbreviations:** PG: pediatric glaucoma; PCG: primary congenital glaucoma; JOAG: juvenile open-angle glaucoma.

## Data Availability

Not applicable.

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
