# Peer review of "Pediatric Glaucoma—From Screening, Early Detection to Management"

_children, 2023, doi:10.3390/children10020181_

Round 1
Reviewer 1 Report
In this manuscript Shen and co-workers review the advances in screening strategies, clinical characteristics, and management of pediatric glaucoma. Overall, this work represents an updated state of the art on these topics, and in my view, it will be interesting for clinicians and people working on the field. However, although the role of genetics in the aetiology and early diagnosis of pediatric glaucoma is increasingly recognized, the section devoted to this issue is not well developed and the manuscript lacks description of classical and recently identified genes (e.g., MYOC, TEK, CPAMD8, etc.). In addition, it is well stablished that loss-of-function of CYP1B1 is the most prevalent genetic alteration and interesting genotype-phenotype correlations, with potential clinical applications, have been described. These points must be addressed to improve the manuscript.
Author Response
Responses to comments from Reviewer 1:
We thank the Reviewer for the valuable comments. We have provided clarification for corresponding changes in our revised manuscript. Listed below are the detailed responses to the comments:
Reviewer 1.
Comments and Suggestions for Authors
In this manuscript Shen and co-workers review the advances in screening strategies, clinical characteristics, and management of pediatric glaucoma. Overall, this work represents an updated state of the art on these topics, and in my view, it will be interesting for clinicians and people working on the field. However, although the role of genetics in the aetiology and early diagnosis of pediatric glaucoma is increasingly recognized, the section devoted to this issue is not well developed and the manuscript lacks description of classical and recently identified genes (e.g., MYOC, TEK, CPAMD8, etc.). In addition, it is well stablished that loss-of-function of CYP1B1 is the most prevalent genetic alteration and interesting genotype-phenotype correlations, with potential clinical applications, have been described. These points must be addressed to improve the manuscript.
Authors’ response:
Thank you very much for the comment. Following the suggestion, we have further described the section on the role of genetics in the etiology and early diagnosis of pediatric glaucoma. The classical and recently identified genes associated with primary or secondary pediatric glaucoma have been described in the revised manuscript. The loss-of-function of CYP1B1 and genotype-phenotype correlations with the potential clinical applications have also been described in the revised manuscript (“etiology” section, page 3-4, lines 86-121).
Reviewer 2 Report
The present review is very extensive and contains almost every important information on pediatric glaucoma. In my opinion still, the manuscript could be improved slightly be looking thoroughly one more time at the structure to remove some redundancies and improve the flow of the presentation/ argumentation.
Major comments:
General
Content-wise, the sections “Patterns of Referral”, “Clinical characteristics” and “Examinations” are sometimes overlapping. I would recommend to think again carefully about how to organize the arguments. Also the “Screening” section often sounds like an “Etiology”-section that is not present in the manuscript.
The structure of the argumentation of the “Improvement of conventional angle surgeries” section should be improved, particularly the part beginning on line 390.
The structure of the section on medical treatment could also be reworked. E.g., you start talking about PGAs (line 532), than go over to beta-blockers etc. to end the paragraphs with information on PGAs again (from line 547).
Examinations:
In 5.4 “Optic Nerve Head Assessment”, you also seem to present facts about macular segmentation (“GCIPL”). This section is a bit confusing.
In general, information on macular OCT/ segmentation for diagnostics and management of PG seems worth mentioning as macular OCT can be helpful to perform e.g., when the optic nerve head is anatomically altered. E.g., look at the results and discussion from Lever et al. 2021 (doi: 10.3390/biology10040260).
OCT can be performed with some children from age 4-5 years old, this can be mentioned to highlight the relevance of handheld-OCT etc. – particularly for children younger than that.
Therapies:
Trabeculotomy is not explained in the section beginning at line 366. Also, it would be nice to explain the difference between goniotomy and trabeculotomy, as both terms are sometimes used as synonyms but the operations refer to different techniques (and approaches e.g., ab interno vs. ab externo).
Minor comments:
Abstract:
Line 11: Who are the authors that contributed equally as first authors?
Line 12: please remove one instance of “correspondence”
Line 14: “Delay” should be Delayed.
Line 15-16: The sentence should be something like: “More effective screening strategies could be beneficial for timely diagnosis and treatment.”
The abstract could be slightly longer. In particular, I think that a short sentence about general therapy of ped. glaucoma would be nice before “Several advanced surgical therapies” (line 16).
Line 19: In the last sentence, the order of the arguments should follow the structure of the paper.
Introduction:
Line 29 (and more): “could” is used a few times across the manuscript, where it should be changed to “can”.
Line 32-35: “prevalence” and “incidence” are used almost like synonyms. Please look at this section carefully and assure to use these terms correctly (“incidence” information requires the mention of the considered period of time).
Line 41-42: “timing of initial disease presentation” sounds redundant.
Line 44-46: In this concluding sentence, the order of the arguments should match the logic structure of the manuscript.
Classification and diagnosis:
Figure 1 is very interesting but in the PDF I received, the resolution wasn’t very high and readability wasn’t optimal.
Screening:
Line 87, 271, 280, 281, 531, 577: “could” should be “can” (cf. comment on line 29).
Line 92: “of these children” can be removed.
Line 148-150: This concluding sentence would work better as an introductory sentence.
Clinical Characteristics:
Line 158: “abnormal corneal irregularity” sounds redundant.
Line 195: “Increase” would be better than “Elongation”.
Examinations:
Line 256: Please explain the “ASOCT” abbreviation as it’s the first time you mention OCT.
Therapies:
Line 449: “hyphemia” should read “hyphema”.
Line 470: the abbreviation “PC” should be “PG”.
Line 489: The use of XEN to improve BGI comes unexpectedly and should be introduced by a short sentence to better guide the reader.
Line 515: “shorter surgery duration” than … not “in”.
Table 1 should be introduced and explained at some point in the “Therapies” section.
Author Response
Responses to comments from Reviewer 2:
We thank the reviewer for the comment regarding our manuscript. We have provided clarification for corresponding changes in our revised manuscript. Listed below are the detailed responses to the comments:
Reviewer 2.
Comments and Suggestions for Authors
The present review is very extensive and contains almost every important information on pediatric glaucoma. In my opinion still, the manuscript could be improved slightly be looking thoroughly one more time at the structure to remove some redundancies and improve the flow of the presentation/ argumentation.
Major comments:
General
Content-wise, the sections “Patterns of Referral”, “Clinical characteristics” and “Examinations” are sometimes overlapping. I would recommend to think again carefully about how to organize the arguments. Also the “Screening” section often sounds like an “Etiology”-section that is not present in the manuscript.
Authors’ response:
We thank the Reviewer for the comment. To avoid overlapping, we have revised the sections “Patterns of Referral”, “Clinical characteristics” and “Examinations” accordingly.
- We have moved the genetic part from the “screening” section to the “etiology” section (page 3-4, lines 86-121).
- Under “Patterns of Referral” section (page 4, lines 144-148), the following sentence is revised: “Elevated IOP (> 21 mmHg) and external abnormalities (e.g., corneal opacity, enlarged eyeballs, tearing, and photophobia) were the main referral reasons. Despite this, healthcare professionals or general ophthalmologists should remain vigilant in evaluating relevant signs, especially for children with visual anomalies or systematic diseases known to be associated with glaucoma,”
- Under “Anterior Segment Abnormalities” section (page 4, lines 165-167), the following sentence is revised: “Increased corneal diameter (i.e., buphthalmos) is a significant corneal abnormality to be recognized in pediatric glaucoma and usually occurs in young children before the age of three.”
- Under “Posterior Segment Changes” section (page 5-6, lines 198-209), the following sentence is revised: “Assessment of the optic nerve head (ONH) is crucial for the diagnosis and monitoring of pediatric glaucoma. Abnormalities of the optic disc, including increasing or increased cup-disc ratio (CDR), CDR asymmetry, and focal thinning, form one of the criteria for defining glaucoma or glaucoma suspect [2]. Of note, CDR parameters (i.e., CDR enlargement or asymmetry) had a high false-positive rate as a referral sign; only 8.5% of children referred with enlarged CDR or CDR asymmetry had confirmed glaucoma [45]. The appearance of optic disc (i.e., ONH cupping reversal) can be improved after considerable IOP reduction in pediatric glaucoma [58, 59]. A higher prevalence of ONH cupping reversal was found in younger eyes [60]. However, some eyes with ONH cupping reversal still experienced continual disease progression after IOP-lowering surgery. This reflects that cupping reversal in pediatric glaucoma may not predict the improvement of the ONH status [59].”
The structure of the argumentation of the “Improvement of conventional angle surgeries” section should be improved, particularly the part beginning on line 390.
Authors’ response:
We thank the Reviewer for the comment. Following the Reviewer’s comment, the argumentation of the “Improvement of conventional angle surgeries” section has been revised (page 9-10, line 404-434).
The structure of the section on medical treatment could also be reworked. E.g., you start talking about PGAs (line 532), than go over to beta-blockers etc. to end the paragraphs with information on PGAs again (from line 547).
Authors’ response:
We thank the Reviewer for the comment. We have removed the description of PGA at the beginning of the medical treatment section and the sentences have been rearranged (“Medications in Pediatric Glaucoma” section, page 15, line 549-573).
Examinations:
In 5.4 “Optic Nerve Head Assessment”, you also seem to present facts about macular segmentation (“GCIPL”). This section is a bit confusing.
In general, information on macular OCT/ segmentation for diagnostics and management of PG seems worth mentioning as macular OCT can be helpful to perform e.g., when the optic nerve head is anatomically altered. E.g., look at the results and discussion from Lever et al. 2021 (doi: 10.3390/biology10040260).
OCT can be performed with some children from age 4-5 years old, this can be mentioned to highlight the relevance of handheld-OCT etc. – particularly for children younger than that.
Authors’ response:
We thank the Reviewer for the comment.
- To avoid confusion, we have renamed the “Optic Nerve Head Assessment” section to “Optical Coherence Tomography (OCT) Assessment” because we have mentioned both the RNFL assessment in the optic nerve head and the GCIPL assessment in the macular region (page 7, line 300).
- We have carefully reviewed the publication by Lever et al. 2021 (doi: 10.3390/biology10040260), and we have mentioned the value of macular OCT in our revised manuscript.
- Under “OCT assessment” section (page 8, line 308-316), the following sentence is revised: “Apart from optic disc OCT assessment, macular OCT has additional value in evaluating retinal layer thickness and visualizing macular morphology [91]. For example, macular OCT can be helpful in measuring retinal thickness if the morphology of optic nerve head is altered by high myopia and axial elongation [91, 92]. Additionally, macular OCT measurements on ganglion cell layer (GCL), inner plexiform layer (IPL), and RNFL reported a high diagnostic accuracy and sensitivity, reflecting its ability to identify glaucomatous eyes and may play a role in glaucoma screening [91]. Therefore, a combination of both optic nerve head and macular segment assessment is recommended in screening and diagnosing PG.”
- We have added the following references:
Lever M, Halfwassen C, Unterlauft JD, Bechrakis NE, Manthey A, Böhm MRR. The Paediatric Glaucoma Diagnostic Ability of Optical Coherence Tomography: A Comparison of Macular Segmentation and Peripapillary Retinal Nerve Fibre Layer Thickness. Biology (Basel). 2021;10(4). Epub 2021/04/04. doi: 10.3390/biology10040260. PubMed PMID: 33805903; PubMed Central PMCID: PMCPMC8064387.
- Shoji T, Sato H, Ishida M, Takeuchi M, Chihara E. Assessment of glaucomatous changes in subjects with high myopia using spectral domain optical coherence tomography. Invest Ophthalmol Vis Sci. 2011;52(2):1098-102. Epub 2010/11/06. doi: 10.1167/iovs.10-5922. PubMed PMID: 21051712.
- We have revised the following sentence under the section “6.4 Optical Coherence Tomography (OCT) Assessment” (Page 8, lines 333-334): “Handheld OCT is feasible in PCG patients without anesthesia or sedation, particularly for children younger than 4-5 years [98].”
References:
- Shah SD, Haq A, Toufeeq S, Tu Z, Edawaji B, Abbott J, et al. Reliability and Recommended Settings for Pediatric Circumpapillary Retinal Nerve Fiber Layer Imaging Using Hand-Held Optical Coherence Tomography. Transl Vis Sci Technol. 2020;9(7):43. Epub 2020/08/25. doi: 10.1167/tvst.9.7.43. PubMed PMID: 32832248; PubMed Central PMCID: PMCPMC7414610.
Therapies:
Trabeculotomy is not explained in the section beginning at line 366. Also, it would be nice to explain the difference between goniotomy and trabeculotomy, as both terms are sometimes used as synonyms but the operations refer to different techniques (and approaches e.g., ab interno vs. ab externo).
Authors’ response:
Thank you for the suggestion. We have now added the explanation of trabeculotomy and goniotomy:
- The following sentence is revised (page 9, lines 396-399): “Trabeculotomy was first introduced in 1970 as an ab externo procedure [113]. Differ from goniotomy, trabeculotomy is performed by cannulating the Schlemm’s canal from an external method with subsequent centripetal rupture through the trabecular meshwork into the anterior chamber.”
Reference:
- Harms H, Dannheim R. Epicritical consideration of 300 cases of trabeculotomy 'ab externo'. Trans Ophthalmol Soc U K (1962). 1970;89:491-9. Epub 1970/01/01.
Minor comments:
Abstract:
Line 11: Who are the authors that contributed equally as first authors?
Authors’ response:
Thank you for the comments. We have deleted it in our revised manuscript (“Abstract” section, line 11).
Line 12: please remove one instance of “correspondence”
Authors’ response:
Thank you. The repeated “correspondence” has been removed in our revised manuscript (“Abstract” section, line 12).
Line 14: “Delay” should be Delayed.
Authors’ response:
Thank you for pointing out the mistake. We have corrected “Delay” to “Delayed” (“Abstract” section, line 13).
Line 15-16: The sentence should be something like: “More effective screening strategies could be beneficial for timely diagnosis and treatment.”
Authors’ response:
We thank the Reviewer for the comment. We have revised the sentence in our revised manuscript accordingly (“Abstract” section, line 15-16).
The abstract could be slightly longer. In particular, I think that a short sentence about general therapy of ped. glaucoma would be nice before “Several advanced surgical therapies” (line 16).
Authors’ response:
Thank you for the suggestion. We have added more sentences in the abstract (line 14-15). A short sentence regarding general therapy has been added before the sentence “Several advanced surgical therapies” (line 16-21).
Line 19: In the last sentence, the order of the arguments should follow the structure of the paper.
Authors’ response:
We thank the Reviewer for the comment. We have revised the order of the arguments to follow the structure of the paper (page 1, lines 23-24): “Here, we review the classification and diagnosis, etiology, screening, clinical characteristics, examinations, and management of PG.”
Introduction:
Line 29 (and more): “could” is used a few times across the manuscript, where it should be changed to “can”.
Authors’ response:
We thank the Reviewer for the comment. We have changed the “could” to “can” as appropriate.
Line 32-35: “prevalence” and “incidence” are used almost like synonyms. Please look at this section carefully and assure to use these terms correctly (“incidence” information requires the mention of the considered period of time).
Authors’ response:
Thank you for the valuable comment. We have carefully checked the original publications, and all the descriptions in original publications were “incidence”. We have revised and unified the term “incidence” in our revised manuscript.
- Under “Introduction” section (page 2, line 35-40), the following sentence is revised: “The incidence of pediatric glaucoma varies across different populations. For instance, the estimated annual incidence rate of pediatric glaucoma among patients aged < 20 years on presentation is 0.92 per 100,000 populations in Hong Kong [5]. In a series in the United States (US), the reported incidence was 2.29 per 100,000, or 1 per 43,575 residents younger than 20 years [6]. A higher incidence was found in Slovakian gypsies (1/1,250) [7] and Saudi Arabia (1/2,500) [8].”
Line 41-42: “timing of initial disease presentation” sounds redundant.
Authors’ response: We thank the Reviewer for the comment. Following the Reviewer’s suggestion, we deleted “timing of” in our revised manuscript to avoid redundancy (page 2, lines 45-46)
Line 44-46: In this concluding sentence, the order of the arguments should match the logic structure of the manuscript.
Authors’ response:
We thank the Reviewer for the comment. We have revised the order of the arguments to match the logical structure of the manuscript. Under “Introduction” section (page 2, lines 48-50), the following sentence is revised: “This review aims to summarize current evidence regarding classification and diagnosis, etiology, screening, clinical characteristics, examinations, and management of PG, and discuss future challenges and directions in the PG management.”
Classification and diagnosis:
Figure 1 is very interesting but in the PDF I received, the resolution wasn’t very high and readability wasn’t optimal.
Authors’ response:
We thank the Reviewer for the comment. We have replaced figure 1 with a higher resolution version (5120 × 2880 pixels, file size: 1649 KB, MIME type: image/jpeg) in the revised manuscript.
Screening:
Line 87, 271, 280, 281, 531, 577: “could” should be “can” (cf. comment on line 29).
Authors’ response:
We thank the Reviewer for the comment. We have changed the “could” into “can” accordingly (lines 151, 351, 360, 361, 670, 719)
Line 92: “of these children” can be removed.
Authors’ response:
Thank you for the comment. we have removed the “of these children” in our revised manuscript (line 156).
Line 148-150: This concluding sentence would work better as an introductory sentence.
Authors’ response:
Thank you for the comment. We have moved the concluding sentence to an introductory sentence (“etiology” section, line 87-89).
Clinical Characteristics:
Line 158: “abnormal corneal irregularity” sounds redundant.
Authors’ response:
We thank the Reviewer for the comment. We have carefully checked the original publications (Hu Y et al, 2018), and the term “abnormal corneal irregularity” is in line with the original description. To avoid misunderstanding, we keep the term “abnormal corneal irregularity” in our revised manuscript.
Reference:
Hu Y, Fang L, Guo X, Yang X, Chen W, Ding X, et al. Corneal Configurations and High-order Aberrations in Primary Congenital Glaucoma. J Glaucoma. 2018;27(12):1112-8. Epub 2018/09/05. doi: 10.1097/ijg.0000000000001049.
Line 195: “Increase” would be better than “Elongation”.
Authors’ response:
We thank the Reviewer for the comment. Following the Reviewers’ comment, we have replaced “Elongation” with “Increase” (“Change in Axial Length and Refraction” section, page 6, line 211).
Examinations:
Line 256: Please explain the “ASOCT” abbreviation as it’s the first time you mention OCT.
Authors’ response:
Thank you for pointing out the mistake. We have added the full term paragraph (“anterior segment optical coherence tomography (ASOCT)”; page 7, lines 272-273).
Therapies:
Line 449: “hyphemia” should read “hyphema”.
Line 470: the abbreviation “PC” should be “PG”.
Line 515: “shorter surgery duration” than … not “in”.
Authors’ response:
Thank you very much for point them out. We have made above suggested changes accordingly.
Line 489: The use of XEN to improve BGI comes unexpectedly and should be introduced by a short sentence to better guide the reader.
Authors’ response:
We thank the Reviewer for the comment. We have introduced the use of XEN to improve BGI by a short sentence (page 12, lines 510-514): “Since the potential complications of BGI include early hypotony and late corneal de-compensation, the combination of the XEN gel stent inserted ab externo in the anterior chamber connected to the Baerveldt tube in the subconjunctival space are expected to overcome these potential complications. Therefore, a new XEN-augmented Baerveldt technique was designed in 2017 [167].”
Table 1 should be introduced and explained at some point in the “Therapies” section.
Authors’ response:
Thank you for the suggestion. We have introduced and explained Table 1 in the “Surgical treatments” section (page 9, line 373-375): “Several clinical studies have been conducted to compare the safety and efficacy of different surgical treatments, including prospective randomized clinical trials. We summarized these comparative studies in Table 1.”
Reviewer 3 Report
excellent review
Author Response
Responses to comments from Reviewer 3:
We thank the reviewer for the comment regarding our manuscript.
Reviewer 3.
Comments and Suggestions for Authors
excellent review
Authors’ response:
We thank the Reviewer for the comment.
Round 2
Reviewer 2 Report
Dear authors,
thank you for the careful revision of your manuscript.
I only have minor last comments to make:
“buphthalmos” (line 165) doesn’t refer only to an increase in corneal diameter above normal, but rather a general enlargement of the globe (also e.g., axial length, scleral thinning) etc.
minor spelling errors.